

# Winter diet and food selection of the Black-necked Crane *Grus nigricollis* in Dashanbao, Yunnan, China

Hao Yan Dong[1,2], Guang Yi Lu[1,2], Xing Yao Zhong[3] and Xiao Jun Yang[1]

[1] State Key Laboratory of Genetic Resources and Evolution, Kunming Institute of Zoology, Chinese Academy of Sciences, Kunming, China
[2] Kunming College of Life Science, University of Chinese Academy of Science, Kunming, China
[3] Administrative Bureau, Dashanbao National Nature Reserve, Zhao Tong, Yunnan, China

## ABSTRACT

The Black-necked Crane *Grus nigricollis* is a globally vulnerable species whose food is the factor determining its long-term survival. Understanding dietary habits, food preferences, and related factors will facilitate the development of effective conservation plans for the protection of this vulnerable species. For this purpose, we used video recordings and sampling of food availability to examine the dietary composition and temporal variation in food selection of Black-necked Cranes wintering in the Dashanbao National Nature Reserve, China. The Black-necked Crane's diet consists primarily of domestic food crops such as grains (74%) and potatoes (8%), in addition to invertebrates (14%). A much smaller proportion of the diet was comprised of turnips and wild herbaceous plants and tubers. There was monthly variation in food selection, partially related to food availability. Grains were most available in November and decreased through the winter, whereas invertebrates were more available in November and February than in December and January. Grain consumption was lowest in November but higher from December through February. Invertebrate consumption was highest in November and February. The cranes preferred to eat grains throughout winter months, while they mainly selected invertebrates in November and February. We suggested invertebrate populations sharply declined in December and January due to the low temperature. In addition, grain consumption was negatively associated with invertebrate availability. In November, when invertebrates were most abundant, and despite a concomitant peak in grain abundance, we suggested cranes exhibited a preference for invertebrates over grains. We recommend that the protection administration provide appropriate supplemental foods for cranes during freezing weather.

Corresponding author
Xiao Jun Yang, yangxj@mail.kiz.ac.cn

## INTRODUCTION

The Black-necked Crane *Grus nigricollis* is a globally vulnerable species, with the main breeding distribution in the high altitude Tibetan-Qinghai Plateau. The cranes migrate short distances to winter in the lower altitude (primarily 2,000–3,200 m) Yunnan–Guizhou Plateau (*Harris & Mirande, 2013*). Through telemetry and banding data, it has become

clear that the birds using the Eastern migratory route (in the following referred to as the Eastern Black-necked Cranes) breed in northern Sichuan and southern Gansu provinces and mainly winter in northeast Yunnan and southwest Guizhou (*Li & Li, 2005*; *Qian et al., 2009*). More than 50% of the wild populations of this species are currently suffering due to significant habitat destruction resulting from grassland degeneration (*Li & Li, 2012*) and conventional agricultural practices that have decreased the diversity of available food types for this species in northeast Yunnan. Food is the factor determining the long-term survival of Black-necked Crane (*Liu et al., 2014*). Thus, understanding the Black-necked Crane's dietary habits, food preferences, and the associated factors will facilitate the development of effective conservation plans for the protection of this vulnerable species.

Determining the dietary composition of wild birds is essential for understanding how the animals interact with their habitats and consequently for identifying their preferred food types (*Baubet, Bonenfant & Brandt, 2004*). Their late discovery and remote range led to a late start in research pertaining to Black-necked Crane's feeding habits (*Harris & Mirande, 2013*). To this point, research surrounding the Black-necked Crane's diet has included quantitative studies on various types of domestic and wild plant foods (*Li & Nie, 1997*; *Bishop & Li, 2001*; *Liu et al., 2014*) and qualitative studies on animal-based foods (*Han, 1995*; *Hu et al., 2002*; *Li & Li, 2005*; *Liu, Yang & Zhu, 2014*). Nonetheless, there remains a lack of synthetic analyses or comparative data regarding the proportions of domestic food crops, animal-matter, and wild plants consumed by the Blacked-necked Crane during the winter.

Until now, fecal microhistological analysis has been the only method used to identify plant material consumed by wintering Black-necked Cranes (*Li & Nie, 1997*; *Liu et al., 2014a*). These studies did not mention the consumption of animal-based foods due to the need for alternative methods to collect this data (*Liu, Yang & Zhu, 2014b*). Generally, fecal analysis can create a bias due to the high variability in digestibility of different food items (*Redpath et al., 2001*). Thus, we chose video recording as an alternative method to better understand the food selection of Black-necked Cranes. This method provided a simple, minimally invasive manner to directly observe the feeding behavior of the threatened bird species in order to estimate their dietary composition (*Newton, 1967*; *Price, 1987*; *Yoshikawa & Osada, 2015*).

Previous studies suggest that variations in temperature may impact food availability (*Kushlan, 1978*; *Stapanian, Smith & Finck, 1999*). As mentioned by *Alonso, Alonso & Bautista (1994)*, low temperatures may decrease grain availability for Common Cranes *Grus grus* by increasing foraging costs due to changes in soil properties. Likewise, temperature is an important correlate of insect activity, further affecting the invertebrate-feeding birds. Higher temperatures are associated with more frequent droughts and dry soils (*Martin, 1985*), while lower temperature cause the soil to freeze. Thus, both affect the degree of insect activity (*McCollogh, Hayes & Bryson, 1927*; *Dowdy, 1937*; *Zhou et al., 2015*) and their availability for birds. Considering this information, we considered that the temperature changes would influence the attributes of available foraging sites, affecting food availability and food selection.

The goal of this research was to better understand factors influencing Black-necked Cranes selection of different feeding habitats during the winter. This information may facilitate the development of strategies to protect the Eastern Black-necked Crane, whose largest population winters in their most important wintering sites in the Dashanbao National Nature Reserve on the Yunnan–Guizhou Plateau (*Li & Yang, 2002*; *Qian et al., 2009*). In this report, we provided a quantitative and comprehensive assessment of the cranes' wintering diet, which included domestic food crops, animal-based foods, and wild plants. We analyzed the cranes' diet composition, food selection, and any correlation between environmental factors, food availability, and food selection.

## METHODS

### Ethics statement

Our research on Black-necked Cranes in Dashanbao National Nature Reserve was approved by the Chinese Wildlife Management Authority and conducted under Law of the People's Republic of China on the Protection of Wildlife (August 28, 2004).

### Field permit

The Administration of ZhaoTong Forestry Bureau approved our study on behavior observation and food availability sampling in the research plot in Dashanbao National Nature Reserve (IDZTL2008163).

### Study site

Dashanbao National Nature Reserve (hereafter referred to as Dashanbao Reserve, 27°18′38″N, 103°14′55″E, altitudes of 3,000–3,200 m), is located in southwest China (Fig. 1), and is listed as a wetland of international significance under the Ramsar Convention on Wetlands. The Dashanbao Reserve is considered an important habitat for Black-necked Cranes, as well as other wintering water birds. It is also known for its upland wetland ecosystem (*Zhong & Dao, 2005*). The study area covers 19,200 ha and is a warm, humid plateau with a monsoon climate characterized by cool, wet summers and cold, dry winters. During winter months, frequent days of sustained freezing temperatures can be expected from December to January. The mean temperature for January is −1 °C, and for July 12.7 °C. The mean annual temperature is 6.2 °C, with 123 frost-free days and 34.6 snow cover days per year. The mean annual precipitation is 1,165 mm (*Li & Zhong, 2010*).

A total of c. 1,200 Black-necked Cranes winter in the Dashanbao Reserve every year, feeding on agricultural farmlands, as well as wild grasslands (*Kong, 2008*). For the purposes of this study, supplemental feeding by humans was ignored because only c. 3 kg of corn are fed to fewer than 50 cranes every day (*Kong et al., 2011a*), which would have little impact on the overall dietary composition and food selection for the cranes. Farmland included fields of cereal (*Avena sativa* and *Fagopyrum tataricum*), potatoes (*Solanum tuberosum*) and turnip (*Brassica rapa* var. *rapa*). Local farming uses a 3-year rotation system, in which cereal is grown one year, followed by two years of potato or turnip, and then back to cereal. Thus, a mosaic of patches of cereal, potato and turnip characterizes the farmland, with each occupying about the same surface area each year. Wild grasslands were comprised of

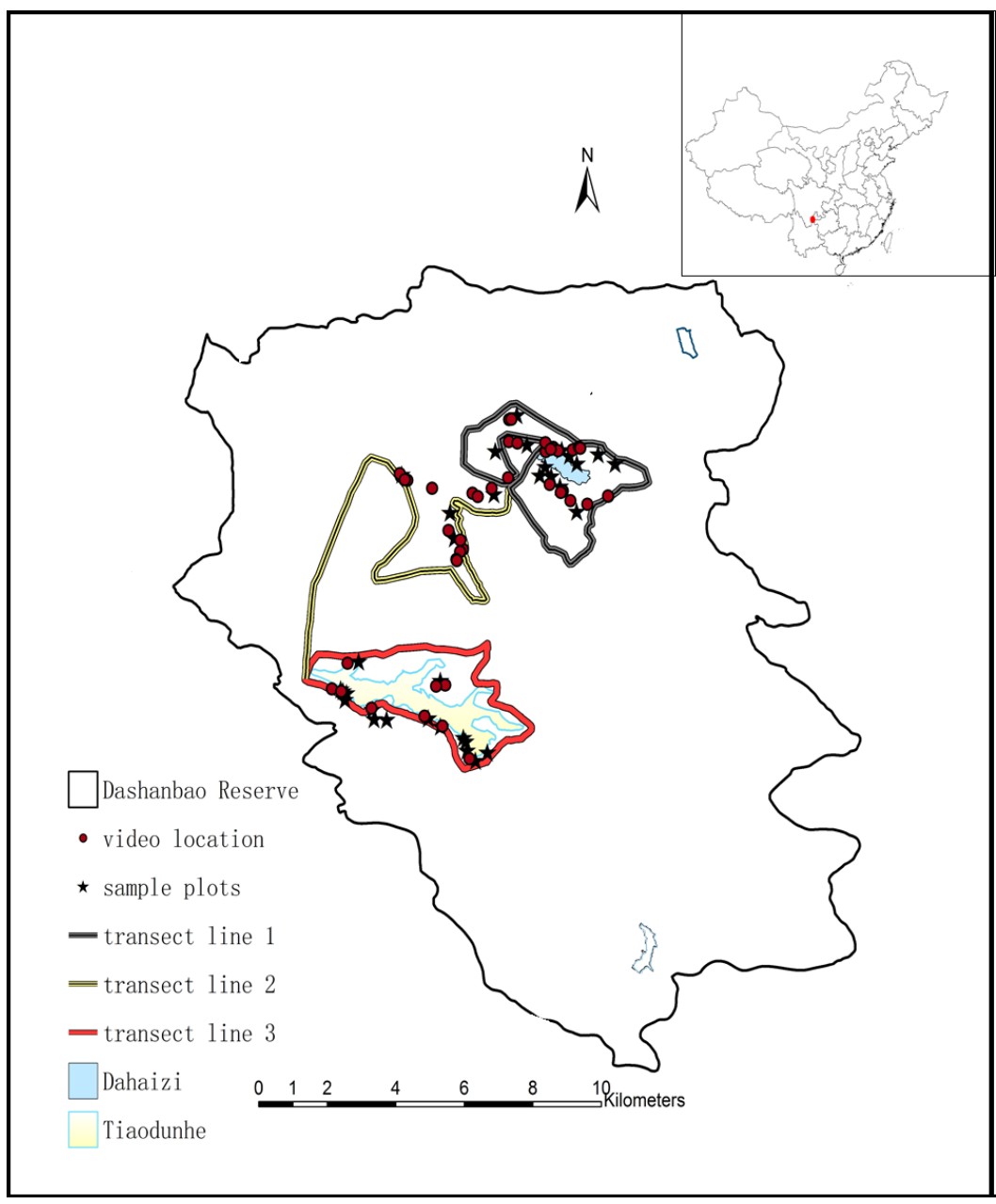

**Figure 1** **Map of the Dashanbao National Nature Reserve (thin black line), showing the location of our study areas.** The red dot at the upper right designates the location of the Reserve within China. Circles indicate sites where we recorded cranes foraging. Stars indicate the sites of food availability sampling. Thick lines indicate the transects. Dahaizi Reservoir (blue) and Tiaodunhe Reservoir (yellow) are indicated. Thin blue lines indicate smaller water bodies.

meadows with minimal water (*Kong et al., 2011a*) and dominated by orchard grass (*Dactylis glomerata*), bluegrass (*Poa annua*), *Leontopodium, Trifolium, Pterospermum heterophyllum, Pedicularis densispica, Luzula multiflora, Hemiphragma heterophyllum* (*Kuang et al., 2008*). The cranes have been reported to forage on *Pedicularis*, *Stellaria*, *Polygonatum* and *Veronica*

(*Kong et al., 2011a*; *Liu et al., 2014a*). The study area covered most of the foraging sites of Black-necked Cranes.

## Bird observations

Field data were collected from November 2013 to February 2015 in the Dashanbao Reserve. Since Black-necked Cranes are highly vigilant and the landscape of the Dashanbao Reserve consists of rolling hills and valleys, we were unable to adequately observe the flocks from our vehicles along the main road in the Dashanbao Reserve and we had to walk along smaller roads. Therefore, we selected three transect routes crossing the mountain ridge of the reserve at two sites which housed the largest flocks of cranes according to the reserve staff's experience and the suggestions from previous research in October 2013 (*Kong et al., 2011a*) (Fig. 1). The majority of cranes arrived in early November and remained feeding in Dashanbao Reserve until early March. We observed cranes for three days every week for 15 weeks between the second week of November and the end of February. In both years, we missed one week due to severe weather.

We videotaped the birds for 5-min intervals each along all transect routes. We walked transects once per day and switched direction of travel on subsequent days. During this time, the cranes were undisturbed and at a maximum distance of 80 m from our point of observation. Based on personal observation, the cranes would startle and flee their feeding site when observed from a distance of less than 60 m. Thus, most sightings were between 60 and 80 m from the birds. For videotaping we chose cranes at random from a within the total number of birds in a flock. This resulted in observations of 50–70% of all cranes in each flock. A total of 505 good quality, 5-min videos were recorded, ensuring sufficient clarity to accurately differentiate among all the consumed food types. For this study, poor quality recordings and those lasting less than 5 min were discarded. A Canon PowerShot SX30 IS digital camera with a 35× optical zoom was used for all the video recordings.

## Foraging behavior

We combined feeding behavior and information about habitat type to determine food type. Food types were classified into 3 categories: (1) domestic crops (including: a. grains, b. potatoes and c. turnips); (2) animal matter (d. invertebrates including primarily earthworms and coleopteran larvae); (3) wild plants (including e. herbaceous plants, f. roots or tubers). Videorecordings of foraging cranes were examined in slow motion to quantify number of pecks per 5-min interval. Every video was watched at least three times to confirm accurate identification of the food types consumed by the feeding crane. Depending on the types of food being eaten, and the peck frequency, four different types of feeding patterns were identified: (1) high pecking frequency and ingestion of all the target food quickly in farmland. This pattern was used primarily for grain on the surface of the ground (see Videos S1 and S2). (2) Digging up the soil to find and consume underground food, such as roots or tubers (including potato and turnip) (see Video S3). Since tubers are too bulky for cranes to swallow, they peck at them repeatedly, swallowing smaller pieces, until the item is completely consumed. This behavior facilitates visual identification of tuber consumption. (3) Consumption of invertebrates is also easily identifiable by a pattern in which the cranes

peck at a plot of turf, capture their prey, and then quickly swallow it (see Videos S4 and S5; Fig. S1). This pattern leaves an obvious disturbance of the turf that can be used for identification (see Figs. S2 and S3). (4) Lastly, the cranes used tugging (*Ellis et al., 1991*), without digging up the soil, primarily for aboveground foods consisting of herbaceous plants. We distinguished this from foraging on grains via a lower pecking frequency and slower swallowing movements (see Video S6, Fig. S4). We recorded the numbers of pecks for each food type. When there was more than one food type in a 5-min recording, we recorded the number of pecks for each type separately.

### Food availability

Given the mosaic landscape of the Dashanbao Reserve, the sampling sites for food availability were selected based on two criteria: (1) The site needed to include a large section of farmland and grassland bordered by farmland with three types of crops in cultivation in the transects. (2) The site must have been selected by at least one flock of cranes for foraging across three transects. Based on these two criteria, twelve plots of farmland (2–6 ha) and twenty plots of grassland (13–43 ha) were selected using Google Earth followed by a field survey (Fig. 1). The proportion of land that each crop and grassland occupied was obtained via monthly sampling. The areas of the sampling sites were calculated using Arcgis 9.2 (ESRI Inc., Redlands, CA, USA).

To investigate the availability of consumable crops, animal matter and wild plants, we proceeded to sample foods using quadrats ($50 \times 50 \times 10$ cm deep) placed at intervals of 100 m along a straight line, guided by GPS localization. We used a direct collection sampling method for cereal grains on unploughed plots and turned the soil for sampling cereal grains on ploughed lands. The latter method was used for sampling potatoes, turnips, invertebrates (e.g., earthworms and Coleoptera larvae), herbaceous plants, as well as tubers within a depth of 10 cm. The length of a crane's bill is 12.4 cm ($n = 10$, 10.5–14.0 cm). We only counted invertebrates larger than approximately 4 mm because that appeared to be the minimal size consumed by the cranes. The count, biomass, and the depth of food types available in each quadrat were recorded. We recorded the depths of frozen soil during the sampling. We placed 176 quadrats in grain fields, and another 222 quadrats in potato and turnip fields in 2013–2015 (sampled monthly for eight months over two years). Earthworms, Coleoptera larvae, herbaceous plants, and roots or tubers were collected from 295 quadrats in grassland in 2013–2015 (eight months). The extracted food items were stored in plastic bags and frozen until processing. After defrosting, cereals, potatoes, turnips, invertebrates, herbaceous plants, and tubers were separated, dried (60 °C, 48 h) and then weighed to determine dry biomass (0.001 g precision).

### Weather variables

Daily temperature values were taken from Zhonghaizi in the Dashanbao Reserve. For our analyses, we used the mean daily temperature and the mean minimum daily temperature. We also counted the number of periods with three or more consecutive days of sustained low temperature (minimum temperature equal to or less than $-10$ °C).These would be days when the ground would remain frozen, thus preventing the cranes from being able to dig for food.

## Statistical analysis

Monthly trophic diversity was estimated using Shannon's diversity index: $H' = -\sum P_i \ln(P_i)$ (*Pielou, 1966*), where $P_i$ represents the proportion of each food type. $P_i = N_i/N$, where $N_i$ is the total number of ingested items of food type $i$ and $N$ is the total number of ingested food items of all types combined. We calculated $H'$ using the proportion estimate derived for each food type present in the sample. We used one-way ANOVA to test differences between months in diversity index. Subsequently, Bonferroni techniques were applied to correct the level of significance of the index. We used the Kruskal–Wallis nonparametric test to explore monthly differences in available biomass of four foods. If the monthly differences were statistically significant, Nemenyi tests in SPSS 20 to test for differences between months. Statistical significance was obtained after applying the Bonferroni correction. We estimated the monthly availability by multiplying the monthly surface of grassland and each type of farmland by the calculated biomass means. Monthly percentage of availability for each food type was calculated by dividing food availability of one food type by the total food availability of all types combined. Food selection by cranes was analyzed using the Savage selectivity index (*Savage, 1931*, cited by *Manly et al., 1993*): $W_i = O_i/\pi_i$, where $O_i$ is the proportion of the sample of used resource units that are in category $i$, and $\pi_i$ is the proportion of available resource units that are in category $i$. The proportion $O_i$ is calculated using the formula $O_i = u_i/u_+$, where $u_i$ represents the number of ingested items of a specific food type and $u_+$ represents the total number of ingested items of all food types. Likewise, the proportion of mean available biomass for a food type is calculated using the formula $\pi_i = A_i/A_+$, with $A_i$ representing the biomass of available resource in category $i$, and $A_+$ the biomass of all available resource units (*Manly et al., 1993*; *Avilés, Sánchez & Parejo, 2002*). This Savage selectivity index can range from 0 to infinity, with 0 indicating maximum negative selection, 1 indicating no selection bias and infinity indicating maximum positive selection (*Manly et al., 1993*). The statistical significance of the selection for each food type from a distribution proportional to its availability was tested using the statistic $(W_i - 1)^2/\text{s.e}\,(W_i)^2$ (*Manly et al., 1993*), which follows the critical value of a $\chi^2$ distribution with one degree of freedom, where s.e. $(W_i)$ is the standard error of $W_i$ calculated using the formula $\sqrt{(1-\pi_i)/(u_+ \times \pi_i)}$ (*Manly et al., 1993*). Statistical significance was obtained after applying the Bonferroni correction for the number of statistical tests (*Rice, 1989*).

We determined the relationships between food availability variables and environmental variability (the mean daily temperatures, minimum daily temperatures, and number of days with frozen soil) using Pearson correlation coefficients in SPSS 20.

To examine the association between food selection and environmental variables, we used Canonical Correlation Analysis (CCA). Multivariate analyses were performed with the software CANOCO (*Ter Braak & Smilauer, 1998*). Preliminary detrended correspondence analysis (DCA) was applied to three food selection datasets (grains, potatoes, invertebrates) to determine the length of the gradient. This DCA revealed that the gradient was greater than 3 standard deviation units (4.2), justifying the use of unimodal ordination techniques (*Ter Braak & Verdonschot, 1995*). Following a preliminary canonical correlation analysis (CCA), we eliminated collinear environmental variables with high variance inflation factors

**Table 1** **Monthly, yearly and two-year combined percentage of food types in the dietary composition of the Black-necked Crane *G. nigricollis* wintering in the Dashanbao National Nature Reserve, China.** For each month in two years, the number of video recordings, the total number of pecks observed, and the percent of pecks directed toward each major food type are shown.

| | | No. of video recordings | No. of pecks | Grain (%) | Potato (%) | Turnip (%) | Invertebrate (%) | Herbaceous plant (%) | Tuber (%) |
|---|---|---|---|---|---|---|---|---|---|
| 2013–2014 | Nov | 46 | 1,180 | 39.73 | 20.29 | 0.01 | 39.05 | 0.93 | 0.00 |
| | Dec | 47 | 1,608 | 95.24 | 1.88 | 0.00 | 1.63 | 1.00 | 0.25 |
| | Jan | 70 | 1,808 | 81.42 | 5.81 | 0.11 | 1.83 | 5.97 | 4.87 |
| | Feb | 50 | 1,212 | 82.51 | 4.95 | 0.00 | 10.56 | 1.49 | 0.50 |
| | Total year | 213 | 5,808 | 74.72 | 8.23 | 0.03 | 13.27 | 2.35 | 1.40 |
| 2014–2015 | Nov | 105 | 1,342 | 49.55 | 11.18 | 0.07 | 33.83 | 1.86 | 3.50 |
| | Dec | 53 | 1,861 | 88.55 | 4.30 | 0.00 | 1.61 | 4.89 | 0.64 |
| | Jan | 66 | 1,495 | 80.27 | 9.03 | 0.07 | 2.34 | 7.16 | 1.14 |
| | Feb | 68 | 1,502 | 73.24 | 5.33 | 0.00 | 20.84 | 0.47 | 0.13 |
| | Total year | 292 | 6,200 | 72.90 | 7.46 | 0.04 | 14.66 | 3.59 | 1.35 |
| Both years combined | | 505 | 12,008 | 73.81 | 7.84 | 0.03 | 13.96 | 2.97 | 1.38 |

(VIF > 20) from further analyses. The variable with the highest significant contribution was included in the analysis (Monte Carlo permutation test $P \leq 0.05$, randomization test with 499 unrestricted permutations). Then, we recalculated the contribution and significance for each variable. Again, the variable with the highest significant contribution was included. This procedure was repeated until none of the variables had a significant contribution. The variables we included were the distributed depths of grain, the depths of potato, the depths of invertebrate, grain availability, potato availability, and invertebrate availability. Canonical correspondence analysis (CCA) with biplot scaling on inter-species distances was used to display the relationship between food selection structure and the 6 environmental variables (main food variables). All multivariate analyses were performed using CANOCO version 4.5 software (*Ter Braak & Smilauer, 2002*).

## RESULTS

### Diet composition

Domestic crops (grains and potatoes) and animal matter (invertebrates) collectively comprised the majority of the Black-necked Crane's diet, followed by wild plants (herbaceous plants, tubers) (Table 1). Turnips comprised less than 1% of the diet on average. When we pooled yearly data, domestic crops and animal matter accounted for 95.61% in total food items, of which grains accounted for 73.81%, potatoes 7.84% and animal matter 13.96%, respectively. In November (both years combined), the proportion of grains consumed was the lowest compared to other months. From December through February, grain consumption was more than twice as high in 2013–2014 and more than 1.4 times as high in 2014–2015. In contrast, the highest consumption of potato and invertebrates occurred in November, followed by January (for potato) and February (for invertebrates), while the lowest consumption for both food types occurred in December. Herbaceous plants and tubers comprised less than 5% of the diet on average. One-way

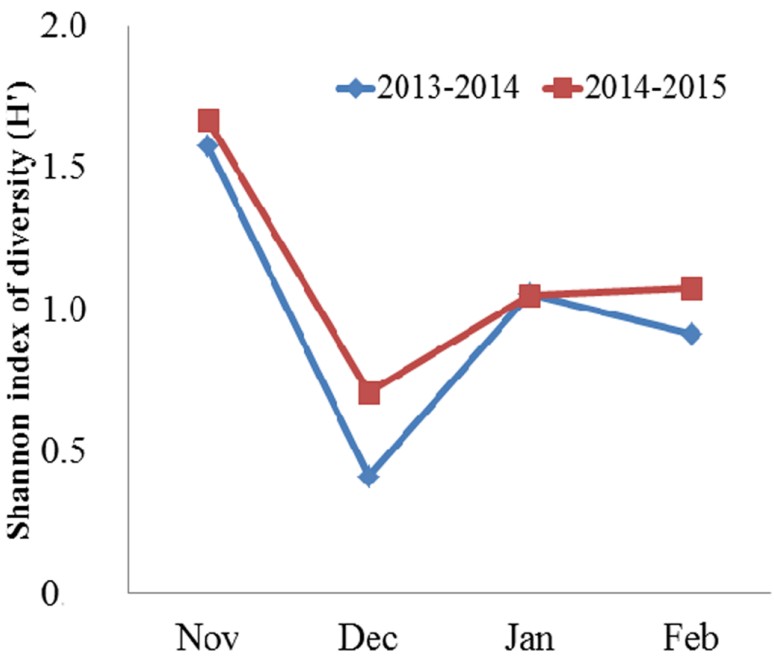

**Figure 2** Monthly mean Shannon index of diversity ($H'$) for the diet of the Black-necked Crane *G. nigricollis* wintering in the Dashanbao National Nature Reserve, China.

ANOVA indicated significant variation on monthly diversity of the diet ($F = 25.00$, $df = 3$, $N = 8$, $P = 0.005$). Then, Bonferroni post-hoc test showed the diversity of the diet was higher in November than in December ($P = 0.006$), January ($P = 0.06$), and February ($P = 0.028$) (Fig. 2).

### Food availability

Wild plant food accounted for the largest proportion of food available in the Black-necked Crane's environment (Table 2). When we pooled yearly data, herbaceous plants and tubers accounted for 89.76%, with 46.51% for herbaceous plants and 43.25% for tubers. Domestic crops (grain 1.17%, potato 1.64%, and turnip 2.94%) and invertebrates (4.48%) accounted for a much lower proportion of total food available. Kruskal–Wallis test indicated significant variation in the monthly availability of grain and invertebrates (grain: $H = 16.402$, $P = 0.001$; invertebrate: $H = 13.081$, $P = 0.004$), whereas we did not find significant effects of month on other types of food ($P > 0.05$). The available biomass of grains in November and December was higher than that in the other two months (Table 2, Nemenyi test, Nov. vs. Dec.: $H = 1.14$, $P = 0.29$; Nov. vs. Jan.: $H = 7.53$, $P = 0.006$; Nov. vs. Feb.: $H = 13.60$, $P = 0.000$; Dec. vs. Jan.: $H = 3.92$, $P = 0.048$; Dec. vs. Feb.: $H = 6.46$, $P = 0.010$). Invertebrate biomass was higher in November and February than that in the other two months (Table 2, Nemenyi test, Nov. vs. Dec.: $H = 7.55$, $P = 0.006$; Nov. vs. Jan.: $H = 4.56$, $P = 0.033$; Nov. vs. Feb.: $H = 0.02$, $P = 0.888$; Feb. vs. Dec.: $H = 8.38$, $P = 0.004$; Feb. vs. Jan.: $H = 5.23$, $P = 0.022$).

**Table 2  Monthly availability of biomass of all food in the Dashanbao National Nature Reserve, China.**

| | | 2013–2014 | | | | | 2014–2015 | | | | | Two year combined |
|---|---|---|---|---|---|---|---|---|---|---|---|---|
| | | Nov | Dec | Jan | Feb | Total year | Nov | Dec | Jan | Feb | Total year | |
| Grains | Mean food biomass (g/0.25 m$^2$) | 0.79 | 0.73 | 0.35 | 0.18 | 0.52 | 2.45 | 0.99 | 0.49 | 0.43 | 1.07 | 0.81 |
| | Sample $N$ | 21 | 23 | 20 | 20 | 84 | 23 | 19 | 20 | 30 | 92 | 176 |
| | Percentage of food availability (%) | 0.45 | 0.54 | 0.38 | 0.26 | 0.41 | 5.08 | 1.17 | 0.56 | 0.93 | 1.94 | 1.17 |
| Potatoes | Mean food biomass (g/0.25 m$^2$) | 2.16 | 1.22 | 0.62 | 0.77 | 1.11 | 1.38 | 1.18 | 0.82 | 0.82 | 1.02 | 1.05 |
| | Sample $N$ | 12 | 14 | 18 | 15 | 59 | 26 | 19 | 34 | 30 | 109 | 168 |
| | Percentage of food availability (%) | 1.07 | 1.16 | 1.08 | 1.56 | 1.22 | 2.51 | 1.81 | 1.50 | 2.47 | 2.07 | 1.64 |
| Turnip | Mean food biomass (g/0.25 m$^2$) | 12.50 | 16.01 | 7.15 | 5.99 | 11.94 | 4.21 | 5.74 | 5.78 | 3.38 | 4.44 | 7.63 |
| | Sample $N$ | 8 | 8 | 4 | 3 | 23 | 8 | 8 | 3 | 12 | 31 | 54 |
| | Percentage of food availability (%) | 2.37 | 5.31 | 3.32 | 2.02 | 3.25 | 2.91 | 3.05 | 2.81 | 1.70 | 2.62 | 2.94 |
| Invertebrate | Mean food biomass (g/0.25 m$^2$) | 1.03 | 0.39 | 0.20 | 0.81 | 0.59 | 0.62 | 0.29 | 0.49 | 0.60 | 0.49 | 0.51 |
| | Sample $N$ | 10 | 12 | 12 | 13 | 47 | 35 | 41 | 41 | 31 | 148 | 195 |
| | Percentage of food availability (%) | 3.77 | 2.54 | 1.74 | 5.65 | 3.43 | 8.23 | 3.05 | 4.56 | 6.30 | 5.54 | 4.48 |
| Herbaceous plant | Mean food biomass (g/0.25 m$^2$) | 7.60 | 3.79 | 1.36 | 3.18 | 4.12 | 2.38 | 2.38 | 2.55 | 2.47 | 2.45 | 2.99 |
| | Sample $N$ | 25 | 22 | 20 | 26 | 93 | 49 | 40 | 51 | 62 | 202 | 295 |
| | Percentage of food availability (%) | 43.53 | 45.05 | 50.73 | 41.4 | 45.18 | 47.11 | 45.41 | 50.42 | 48.45 | 47.85 | 46.51 |
| Tubers | Mean food biomass (g/0.25 m$^2$) | 7.88 | 6.95 | 8.17 | 7.01 | 7.46 | 1.81 | 4.34 | 4.33 | 3.85 | 3.47 | 4.73 |
| | Sample $N$ | 11 | 19 | 19 | 18 | 67 | 41 | 29 | 28 | 47 | 145 | 212 |
| | Percentage of food availability (%) | 48.82 | 45.4 | 42.71 | 49.1 | 46.51 | 34.17 | 45.50 | 40.14 | 40.15 | 39.99 | 43.25 |

## Food selection

In comparing the six types (3 categories) of foods available to the foods selected, the Savage index showed that the cranes preferred grain through the wintering period. Invertebrates were the second preferred food type in November and February. Potatoes were preferred in November in 2013–2014 (Table 3), whereas they were either avoided or showed no significant preference in the other months. Turnips, herbaceous plants and tubers were avoided through the wintering period.

## Relationships between environmental factors and food availability and food selection

We used the Pearson correlation coefficient to determine the correlation between environmental temperatures and the availability of key food items. The results showed the depth distribution of invertebrate was significantly negatively correlated with mean temperature and mean of minimum temperature and was positively correlated with the number of days during which the ground was frozen (Table 4). In addition, the number of invertebrates at depths of 0–1 cm and 1.1–2 cm were positively correlated with temperature and negatively correlated with the number of days with frozen ground (Table 4). The mean depth of the frozen ground was 4.93 cm in December ($n = 10$, 2.6–6.9 cm) and 3.12 cm ($n = 5$, 2.9–3.5 cm) in January (see Fig. S5).

**Table 3  Food selection of Black-necked Cranes (*G. nigricollis*) for the six most available food types in the Dashanbao National Nature Reserve China in relation to month and year.**  Shown is number of video recordings, the number of pecks, food availability ($\pi_i$), food selection ($O_i$), the Savage selectivity index ($W_i$) for each food type, standard error of the index (s.e), and the statistical significance ($P$) for our results. Significance is reached at $P < 0.006$, after applying the Bonferroni correction.

| | | | No. of video recordings | No. of pecks | $O_i$ | $\pi_i$ | $W_i$ | s.e | $P$ | selection |
|---|---|---|---|---|---|---|---|---|---|---|
| 2013–2014 | Nov | Grain | 13 | 470 | 0.40 | 0.04 | 10.34 | 0.15 | * | + |
| | | Potato | 3 | 240 | 0.20 | 0.06 | 3.37 | 0.11 | * | + |
| | | Turnip | 0 | 0 | 0 | 0.23 | 0 | 0.05 | * | − |
| | | Invertebrate | 26 | 462 | 0.39 | 0.02 | 16.32 | 0.19 | * | + |
| | | Herbaceous plants | 4 | 8 | 0.01 | 0.44 | 0.02 | 0.03 | * | − |
| | | Tuber | 0 | 0 | 0 | 0.20 | 0 | 0.06 | * | − |
| | Dec | Grain | 26 | 1,520 | 0.95 | 0.04 | 21.49 | 0.12 | * | + |
| | | Potato | 4 | 30 | 0.02 | 0.04 | 0.42 | 0.12 | * | − |
| | | Turnip | 0 | 0 | 0 | 0.34 | 0 | 0.04 | * | − |
| | | Invertebrate | 7 | 26 | 0.02 | 0.01 | 1.32 | 0.22 | NS | + |
| | | Herbaceous plants | 7 | 21 | 0.01 | 0.22 | 0.06 | 0.05 | * | − |
| | | Tuber | 3 | 11 | 0.01 | 0.35 | 0.02 | 0.03 | * | − |
| | Jan | Grain | 22 | 1,472 | 0.81 | 0.03 | 26.71 | 0.13 | * | + |
| | | Potato | 13 | 105 | 0.06 | 0.05 | 1.21 | 0.10 | NS | + |
| | | Turnip | 2 | 2 | 0 | 0.12 | 0.01 | 0.06 | * | − |
| | | Invertebrate | 12 | 33 | 0.02 | 0.01 | 1.77 | 0.23 | * | + |
| | | Herbaceous plants | 6 | 108 | 0.06 | 0.11 | 0.53 | 0.07 | * | − |
| | | Tuber | 16 | 88 | 0.05 | 0.67 | 0.07 | 0.02 | * | − |
| | Feb | Grain | 22 | 1,000 | 0.83 | 0.01 | 57.09 | 0.24 | * | + |
| | | Potato | 5 | 60 | 0.05 | 0.05 | 1.08 | 0.13 | NS | + |
| | | Turnip | 0 | 0 | 0 | 0.07 | 0 | 0.10 | * | − |
| | | Invertebrate | 14 | 128 | 0.11 | 0.04 | 2.54 | 0.14 | * | + |
| | | Herbaceous plants | 5 | 18 | 0.01 | 0.33 | 0.05 | 0.04 | * | − |
| | | Tuber | 4 | 6 | 0 | 0.50 | 0.01 | 0.03 | * | − |
| 2014–2015 | Nov | Grain | 18 | 665 | 0.50 | 0.17 | 2.97 | 0.06 | * | + |
| | | Potato | 18 | 150 | 0.11 | 0.11 | 1.05 | 0.08 | NS | + |
| | | Turnip | 6 | 1 | 0 | 0.10 | 0.01 | 0.08 | * | − |
| | | Invertebrate | 41 | 454 | 0.34 | 0.06 | 5.32 | 0.10 | * | + |
| | | Herbaceous plants | 14 | 25 | 0.02 | 0.34 | 0.05 | 0.04 | * | − |
| | | Tuber | 8 | 47 | 0.04 | 0.22 | 0.16 | 0.05 | * | − |
| | Dec | Grain | 20 | 1,648 | 0.89 | 0.06 | 15.07 | 0.09 | * | + |
| | | Potato | 10 | 80 | 0.04 | 0.07 | 0.62 | 0.08 | * | − |
| | | Turnip | 0 | 0 | 0 | 0.14 | 0 | 0.06 | * | − |
| | | Invertebrate | 8 | 30 | 0.02 | 0.04 | 0.43 | 0.12 | * | − |
| | | Herbaceous plants | 11 | 91 | 0.05 | 0.30 | 0.16 | 0.04 | * | − |
| | | Tuber | 4 | 12 | 0.01 | 0.39 | 0.02 | 0.03 | * | − |
| | Jan | Grain | 22 | 1,200 | 0.8 | 0.03 | 26.79 | 0.15 | * | + |
| | | Potato | 19 | 135 | 0.09 | 0.09 | 1.06 | 0.08 | NS | + |

**Table 3** (*continued*)

|  |  | No. of video recordings | No. of pecks | $O_i$ | $\pi_i$ | $W_i$ | s.e | $P$ | selection |
|---|---|---|---|---|---|---|---|---|---|
|  | Turnip | 3 | 1 | 0 | 0.05 | 0.01 | 0.11 | * | − |
|  | Invertebrate | 10 | 35 | 0.02 | 0.06 | 0.38 | 0.10 | * | − |
|  | Herbaceous plants | 10 | 107 | 0.07 | 0.40 | 0.18 | 0.03 | * | − |
|  | Tuber | 2 | 17 | 0.01 | 0.37 | 0.03 | 0.03 | * | − |
| Feb | Grain | 26 | 1,100 | 0.73 | 0.03 | 24.41 | 0.15 | * | + |
|  | Potato | 7 | 80 | 0.05 | 0.06 | 0.93 | 0.10 | NS | − |
|  | Turnip | 0 | 0 | 0 | 0.09 | 0 | 0.08 | * | − |
|  | Invertebrate | 28 | 313 | 0.21 | 0.04 | 4.79 | 0.12 | * | + |
|  | Herbaceous plants | 5 | 7 | 0 | 0.36 | 0.01 | 0.03 | * | − |
|  | Tuber | 2 | 2 | 0 | 0.42 | 0 | 0.03 | * | − |

Notes.
NS, $P > 0.006$; +, Positive selection; −, Negative selection.

**Table 4** Pearson correlations between the environmental variables and invertebrate food variables for Black-necked cranes (*G. nigricollis*) in the Dashanbao National Nature Reserve China.

|  |  | Invertebrate depth | Invertebrate numbers (0–1 cm) | Invertebrate numbers (1.1–2 cm) |
|---|---|---|---|---|
| Mean temperature | Correlation coefficients | −0.721[*] | 0.740[*] | 0.690 |
|  | $P$-value | 0.043 | 0.036 | 0.058 |
| Minimum temperature | Correlation coefficients | −0.730[*] | 0.843[**] | 0.775[*] |
|  | $P$-value | 0.040 | 0.009 | 0.024 |
| Number of days with frozen ground | Correlation coefficients | 0.779[*] | −0.842[**] | −0.797[*] |
|  | $P$-value | 0.023 | 0.009 | 0.018 |

Notes.
[*]Correlation is significant at the 0.05 level (2-tailed).
[**]Correlation is significant at the 0.01 level (2-tailed).

Canonical correspondence analysis (CCA) exhibited the relationship between environmental factors and grain selection, potato selection and invertebrate selection in different patterns (Fig. 3). The eigenvalues for the first two axes in Fig. 3 were 0.223 and 0.007, respectively. The food selection-environmental correlations for the first two axes were 0.986 and 0.714, respectively. The first two axes of the CCA explained 96.6% of the total variance in food selection data and food variables, of which 93.8% was contributed by the first axis, and 2.88% by the second axis. Invertebrate availability (0.77), potato availability (0.65) and grain availability (0.53) were positively associated with the first axis, while the distributed depths of invertebrate (−0.78), the depths of potato (−0.68) and the depths of grain (−0.49) were negatively associated with the first axis. CCA axes 1 and 2 separated the food selection into groups for grain selection, potato selection and invertebrate selection. Invertebrate selection was positively associated with invertebrate availability and was negatively associated with the invertebrate depths. Potato selection was negatively associated with invertebrate depths. Grain selection was positively associated with invertebrate depth, followed by potato and grain depths, which were negatively
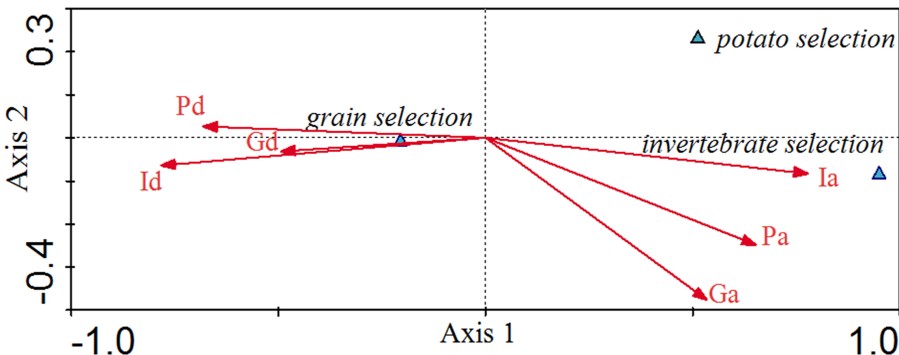

**Figure 3 Canonical Correlation Analysis (CCA) showing the relationship between environmental variables and selection for grain, potato, and invertebrates.** Environmental variables are represented by arrows and their abbreviation: Ia, Invertebrate availability; Pa, Potato availability; Ga, Gain availability; Id, Invertebrate depth; Gd, Grain depth; Pd, Potato depth. The *x*-axis refers to Axis 1 and the *y*-axis refers to Axis 2.

associated with invertebrate availability. As invertebrate availability seasonally decreased, cranes increased their grain consumption.

## DISCUSSION

### Diet composition

The variation in diet of the Black-necked Cranes was systematically studied for the first time using video recording. The results revealed that the wintering diet of the Black-necked Crane in the Dashanbao Reserve mainly consisted of domestic crops (e.g., grains and potatoes), and invertebrate animals. Turnips and wild plants foods (such as herbaceous plants, and tubers) accounted for a much lower proportion of their diet. These results are similar to those of a previous report in which fecal analysis was used to study the crop and wild plant consumption of a subpopulation of Black-necked Cranes wintering at the Yarlung Zangpo Valley National Natural Reserve. However, the report on the cranes in the Yarlung Zangpo Valley National Natural Reserve did not calculate the proportion of animal-based food (*Bishop & Li, 2001*). It is important to note that initial estimates approximated that 13.96% of the Dashanbao Black-necked Crane's diet would consist of invertebrates. In comparison, animal matter comprises less than 10% of the diet for Common Cranes in the Holm Oak Dehesas (*Avilés, Sánchez & Parejo, 2002*), and 2–3% of the diet for various crane species in different regions of the world (*Irene, 1980*; *Reinecke & Krapu, 1986*). Certain crane species feed primarily on animal matter while wintering in some sites. These include, the Lesser Sandhill Crane (*G. canadensis canadensis*) (*Davis & Vohs, 1993*), Whooping Crane (*G. americana*) (*Pugesek, Baldwin & Stehn, 2013*), and Red-crowned Crane (*G. japonensis*) (*Li et al., 2014*). Demoiselle Cranes (*Anthropoides virgo*) (*Sarwar et al., 2013*), Florida Sandhill Cranes (*G. c. pratensis*) (*Rucker, 1992*) and Common Cranes show similar preferences for invertebrates (*Avilés, Sánchez & Parejo, 2002*).

Current research on the proportion of animal-based foods in the diet of Black-necked Cranes has solely focused on describing species (*Han, 1995*; *Hu et al., 2002*; *Li & Li, 2005*; *Liu, Yang & Zhu, 2014b*). Thus, there is a need for additional quantitative investigations

into the Black-necked Cranes feeding habits, including invertebrate consumption. Likewise, more data are needed to study the feeding habits of Black-necked Cranes over a greater distribution of locations. This would greatly enhance our understanding of the dietary habits of this species.

Previous studies using fecal analysis to assess the proportion of the mentioned food categories in the Black-necked Crane's diet have produced results inconsistent with our study. These studies largely reported a wild plant diet (leaves, roots and tubers), while failing to mention the inclusion of domestic crops or invertebrates in the diet of cranes in the Dashanbao Reserve (*Liu et al., 2014a*). This inconsistency has two possible explanations: the method to analyze the data and the sampling procedures. First, different methods were used to analyze the diet. With fecal analysis, wild plant fiber may therefore have been easier to detect in feces than the potato and grain fibers or invertebrate larvae residues, despite the latter two making up a larger proportion of the diet. *Liu et al. (2014a)* mentioned potato cuticles were not detectable in the fecal sample of a crane that due to the digestibility of the food type. With video observation, we were able to directly estimate the frequency on which a particular food type was fed on, without concern for variations in digestibility. While, video observation enables the detection of even highly digestible food, it is often more difficult to identify the specific food types that are seen consumed. Thus, it requires more careful observation and detection of feeding patterns to identify food items. This may also be seen as an advantage, as it can provide us with more complete foraging information, including actual foraging behavior. We are thus able to successfully estimate the digestible compositions of a birds' diet (*Robinson & Holmes, 1982*; *Rundle, 1982*). Second, our results infer that the sampling time may have greater impact on identifying food types which change with monthly variations. For example, as a climate-restricted food, invertebrates are difficult for Black-necked Cranes to find in December and January (Table 2) (see below discussion). Fecal analysis of Black-necked Crane's diet in the previous study did not mention sampling time in Dashanbao Reserve (*Liu et al., 2014a*). It is possible that different sampling times caused the difference from our results.

## Monthly variation and diet selection

In November, a high proportion of the Black-necked Crane's diet consisted of domestic crops (principally grain) and invertebrate organisms (Table 1). This may be because the availability of those food types was the highest immediately after the birds arrived (November). The birds require a balanced diet, including a variety of nutrients from different food types. In November when both grains and invertebrates were most available, invertebrates were consumed more than at any other time. In contrast, grains were consumed less than in other months. This suggests that the cranes likely prefer invertebrates over grains, potentially because invertebrate organisms provide a greater source of protein and calcium than available in grains. These nutrients are essential for their migration fitness and overall survival. Cranes consumed only a minimal quantity of wild plants despite their larger proportion of available biomass as compared to that of domestic crops and animal matter (Table 2). It is possible that cranes prefer domestic crops or animal matter over wild plants because (1) herbaceous plants may have lower caloric content than grains or

animal matter; (2) there is insufficient density of vegetation suitable for the cranes to forage preferred species such as *Pedicularis*, *Stellaria*, *Polygonatum* and *Veronica* (*Kong et al., 2011*; *Liu et al., 2014a*).

## Environmental factors compared to food selection

Based on the results of our CCA, the grain selection and invertebrate selection present two different patterns. Grain selection was positively correlated with invertebrate depth and negatively correlated with invertebrate availability. However, invertebrate selection shows the opposite pattern. Falling temperatures and freezing soils reduced the availability of invertebrates and increased the depth of invertebrates, especially for December and January (Table 4). Therefore, cranes primarily fed on grains during December and January and fed on invertebrate animals in November and February. Potato selection was negatively associated with the depths of invertebrates. When invertebrates are at increased depth due to low temperatures, the cost of digging for potatoes also increases as a result of frozen soil.

## Management implications

Our results support previous reports that Black-necked Cranes generally prefer farmlands, and avoid grasslands (*Kong et al., 2011a*), likely due to the availability of domestic crops and invertebrates to feed on, as well as other habitat features. We agree with Kong's views (*2011*) that higher quantities and densities of food as well as looser soil structure in farmlands facilitate food collection by the cranes. During colder weather (December or January), the invertebrate shortage is exacerbated. We recommend that the protection administration should supplement additional foods for cranes during the cold-weather periods, and restore grassland foraging habitat. This would support the cranes' need for dietary diversity and would benefit the farmers by reducing economic losses resulting from the cranes feeding on newly planted crop seeds during their late spring migration (in March). To further ease the conflict between cranes and local farmers, it is advisable to cultivate crops in a certain area that may be left unharvested for the cranes to eat. Furthermore, it is necessary to maintain adequate traditional croplands to sustain this vulnerable species, as many of these conventional cultivations (grains, potatoes and turnips) have been replaced by more economic crops (*Lepidium meyenii* Walp) in the Dashanbao Reserve.

# ACKNOWLEDGEMENTS

We thank Shimei Li and Yuanjian Zhen for their help in our field work, and staff of Dashanbao National Nature Reserve for their valuable support in the field.

## Funding

The study was supported by the ICF (International Crane Foundation). The funders had no role in study design, data collection and analysis, decision to publish, or preparation of the manuscript.

## Grant Disclosures

The following grant information was disclosed by the authors:
ICF (International Crane Foundation).

## Competing Interests

The authors declare there are no competing interests.

## Author Contributions

- Hao Yan Dong conceived and designed the experiments, performed the experiments, analyzed the data, contributed reagents/materials/analysis tools, wrote the paper, prepared figures and/or tables, reviewed drafts of the paper.
- Guang Yi Lu performed the experiments, analyzed the data, reviewed drafts of the paper.
- Xing Yao Zhong performed the experiments, contributed reagents/materials/analysis tools.
- Xiao Jun Yang conceived and designed the experiments, performed the experiments, analyzed the data, contributed reagents/materials/analysis tools, reviewed drafts of the paper.

## Ethics

The following information was supplied relating to ethical approvals (i.e., approving body and any reference numbers):

Our research on Black-necked Cranes in Dashanbao National Nature Reserve was approved by the Chinese Wildlife Management Authority and conducted under Law of the People's Republic of China on the Protection of Wildlife (August 28, 2004).

## Field Study Permissions

The following information was supplied relating to field study approvals (i.e., approving body and any reference numbers):

The Administration of ZhaoTong Forestry Bureau approved our study on behavior observation and sampling collection in the research plot in Dashanbao National Nature Reserve (IDZTL2008163).

## Data Availability

The raw data was supplied as Supplemental Information.

## Supplemental Information

Supplemental information for this article can be found online at http://dx.doi.org/10.7717/peerj.1968#supplemental-information.

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
