# Peer review of "Winter diet and food selection of the Black-necked Crane Grus nigricollis in Dashanbao, Yunnan, China"

_PeerJ, doi:10.7717/peerj.1968_

## Round 0.1 · original submission · Major Revisions

Overview

This study used video analysis to examine the diet of black-necked cranes from November to February in two years on their wintering grounds in China. The authors examined availability of the different foods to assess selectivity and examined the relationship of diet and food availability to temperature. Both reviewers found the manuscript a valid and useful contribution and recommended acceptance.

One reviewer provided an annotated manuscript file with several suggested minor revisions. However, in reading the manuscript, I identified many problems with the clarity and completeness of the presentation and some possible problems with the statistical analysis. I am therefore providing detailed feedback and recommending major revisions. It is likely that the manuscript will need to be reviewed after the changes, possibly including an evaluation of the statistical analysis. Many specific comments on grammar and style are provided on an annotated manuscript. I used the version already annotated by the first reviewer to facilitate the authors' review. I used yellow highlights to identify problems. I used inserted comments to explain the problem or to suggest alternative wording or punctuation. I did not provide such explanations for all the problems because they were very numerous and it is likely that the text will change substantially. When a problem was identified in one area, the author needs to search for other cases because I may not have marked them all.

Introduction
L42ff. Your manuscript is about a wintering population. I assume that they do not breed in this area. It is not clear if your statement about the effect of food availability is related only to the wintering population. You need a reference for it being a vulnerable species and for food selectivity as one of the most important problems. Is the problem really food selectivity or is it food availability? I have a problem seeing how selectivity can be a cause of vulnerability except for very specialized species.
L81 mentions Eastern Black-necked Cranes without providing any previous information about sub-population differentiation.
L82. Does referring to the study area as a stopover site imply that the cranes do not remain there for the winter but continue on to another location? If so, how could you study them for 4 months? (It is possible that the definition of 'stopover' includes wintering habitat, but that is not the way I have heard the term used.)

Methods
Figure 1 provides much useful information, but readers may miss that because you did not indicate that the transects and food sampling sites were shown on Figure 1 when you introduced these aspects of your methods.

L113-114, 138-140. The habitat and foods are incompletely described. Please be more specific about what species are included in 'cereal' and provide scientific names for all species when first mentioned. Slightly more information might be useful for 'bastard speedwell'. I was not familiar with this plant and could not find much information about it as a food on the internet. Indeed, there appeared to be a couple of different species with the same common common name. Is the plant a food crop for humans? Is it an annual with only crop residue left in the fields? What parts do cranes eat? Similarly with potatoes, are the cranes eating small or damaged tubers left after harvest or some other part of the plant? For the grasslands, is there any botanical description of the dominant plant species available? I checked Kong et al. 2011a but did not find a good description of the flora there either. For invertebrates, please list at least the major taxa. The terms 'worms' and 'grubs' are not technical biological terms. Do you mean earthworms and beetle larvae? If so, do you have any idea what species are involved?

L138ff. The food types are not presented in a consistent manner. L138 refers to 3 categories of food and within parentheses lists the types of food in those categories. L236 refers to 6 types. Fig. 3 has 7 types. One of the problems is that it is not always clear when you are combining or separating categories, e.g. are 'roots and tubers', 'leaves and herbaceous plants' considered one or two food types. Furthermore, it is not clear how leaves and herbaceous plants differ because herbaceous plants usually have leaves.

L120 ff. The field video recording procedure is not clear. What were the start and end dates in each year? You present data for 4 months but state that your study lasted only 15 weeks, so I presume that it did not include all of November and February. I cannot understand the sampling procedure. This is critical to know whether your sample of feeding cranes was appropriately random in relation to habitat. The role of the transect routes is unclear. Are you implying that you walked the same transect routes each day? When you found cranes, how did you decide whether to film or not? To select individuals within a group, did you use a truly randomly procedure or only select haphazardly? If random, describe briefly the procedure. How many individuals did you record from the same foraging group before moving on? The recording distances are also unclear and appear contradictory. You say that the distances were less than 80 m and usually 30 - 60 m, but you also say that the cranes flushed at less than 60 m. In selecting video sequences for use, is it possible that using only videos in which the food type was clear could exclude some food types more than others?

L144 ff. You need to have a much more explicit description of how you identified and recorded the food types from the videos. Did you use only the behavioral descriptions from this section or did you combine behavior with information about habitat type in determining food type? Very importantly, it is not clear at what time scale you identified the food types. For example, was each 5-min recording classified as a single food type or did you count each peck? In some of your videos, there are many pecks before a potato is fully consumed. Were those all those pecks counted separately or as a single event? How did you deal with recordings in which more than one type of food was eaten? This has implications for selectivity if food use was based on pecks and food availability was based on dry weight.

The descriptions of feeding behavior need much clarification. 'Hauling and dragging' are not clear terms to describe a bird feeding. Ideally, you should search for previous researchers who have described similar feeding behavior and use the same terminology, if appropriate. I did a short search to see if I could find better terminology. I found a reference to a conference proceeding by Ellis et al that appeared to describe feeding movements in cranes. Unfortunately, I could not find an electronic version. I contacted Dr. Ellis, and he kindly sent me a scanned copy (and also a related article on social movements). I will include pdfs of these articles in my reply. Figure 2 does not seem to add much clarity to the feeding descriptions because it shows only a single moment when the beak is in contact with the food item; it is not clear what information you intend to convey with this drawing. Consider either improving or deleting this drawing. Some of the still photographs are similarly not very revealing. The videos are more enlightening. Are they the highest resolution possible or would it be possible to provide higher resolution to see the food items more clearly? These videos would be much more useful if the descriptions were much more explicit. For example, "Video S1 shows 2 cranes foraging in a potato field on (date and time). They peck at small items, probably invertebrates. At 00:31 the crane on the right starts to feed on a potato tuber. At 01:40 the potato is taken by the second crane which feeds on it for the rest of the sequence, swallowing two large pieces." (My description may not be accurate, but should give you an idea of what I mean by a detailed description to help the reader understand what you are trying to convey.) The videos and figures should be numbered in the order in which they are referred to in the manuscript. Note that the caption for video S6 does not appear to match the caption at all.

L159ff. Your determination of food availability is also not clear. If I understand correctly, the food availability sampling locations did not correspond to the transects on which you recorded feeding. Is that correct? If so, have you considered whether selectivity is due to a difference between the availability that you measured and the availability that the birds that you observed experienced? You should start with a clear, general statement about how you assessed food availability. It seems that you first assessed the proportion of the area devoted to each crop and to grassland and then assessed the food density on each type using quadrats spaced regularly along a transect. Until you explain this, readers cannot understand the relevance of the section called Sampling area. On L160, your description of sites on a boundary between grassland and cropland is not clear because you go on to describe a choice of sites in each area separately. (Perhaps the problem is a misunderstanding of the word 'boundary' as also occurred in the key to Fig. 1.) On L172-173, there appears to be an error because you describe two methods but attribute both to unploughed land. You need to introduce the concept of the quadrats earlier in the paragraph when you refer to the transects, because it is not clear what quadrats are involved when you refer to them on L175. You need to be more specific about what invertebrates you collected from the samples and which ones you did not include (presumably there were many very small invertebrates that you did not bother to count). L184 refers to the cropland. Was grassland also included in a similar area calculation? In your data files, I see data for biomass for crops but not for grassland. I also do not see the calculation for total availability.

You present data for food depth distribution and depth of freezing, but I was unable to find a description of the related procedures in the Methods.

Statistical analysis. The description of your statistical analysis is also unclear and may not be correct. It is not clear how you could calculate Shannon's diversity for each food type; it measures the relative contribution of all food types. Your description of your calculation of the Savage index does not correspond the description in Resource Selection Ch. 4 by Manly and neither does the definition of SE used to calculate the significance of the difference. Whether this is an error of writing or statistics, is not possible to determine from the manuscript. In addition, you do not appear to have corrected for multiple testing either in the calculation of the significance of your index values nor in the large correlation matrix relating food availability to environmental conditions. (Manly does refer to this problem in Ch. 4.) It is critical that you check your calculations and your description and also consult with a statistician to be sure everything was correctly done. Also, your Methods section implies that you only examined the correlation between food availability and environmental conditions but you also carried out all the correlations within each of these data sets. You need to consider carefully what questions you want to ask and provide the corresponding analyses. Most contemporary researchers would not present a huge correlation matrix but rather would carry out a multivariate analysis. You present seasonal patterns in food availability but no analysis to show whether the seasonal differences were statistically significant. While it is possible to talk about general trends, statistical support might be useful if you consider the seasonal patterns to be potentially important.

Results
Your results for the proportion of different food types in the diet would be clearer in a table than in Figure 3 because the lesser-used types are very difficult to determine from the figure. I also recommend that you provide an average proportion for each year as well as an average for both years combined. This will give readers a basis for understanding each food item overall and provide support for the summary values you give in the Abstract. The table could also give sample size (number of recordings) for each month. It should indicate the units of measurement (proportion of pecks, proportion of recordings, or something else). You would need to be careful in calculating the mean. Presumably, the mean of the 4 months would be more appropriate than the mean of all samples if the sample size differs among months. These same concerns (table vs. figure, statistics between months, yearly and overall means) apply to the presentation of your diversity index data and your food availability data. If you do not provide the proportion of use and of available food in tables, then the table in which you present selectivity (Table 1 in the present version) needs to have the proportion of use and the proportion of availability as well as the selectivity for the reader to clearly understand the pattern.

L246. Consider switching the figure to a correlation pattern, x = invertebrate availability, y = invertebrate use.

In general, when you present data in figures and tables, please use a consistent order that matches the presentation in your methods, so that readers will be able to follow the patterns easily. Also, please be cautious in using abbreviations. Table 2, for example, is almost impossible for readers because of all the abbreviations, most of which are non-standard. In this case, it would be worth using whole words as much as possible. You should avoid narrow columns that result in words being divided arbitrarily (e.g., tuber in Table 1). When you have percent on the vertical axis, you should not put a % symbol after each value but indicate in the axis label that the units are percent, e.g., 'Available foods (% dry mass)'. You should not show a continuous line between the two years because you are missing data for 8 months; instead, break the line between years.

Discussion
The discussion could use a tighter organization so that for each of your findings you present how your results differ or are similar to previous research and then the explanation for any differences and the importance of the results. You have most of the relevant information, but it is not organized for maximum clarity.
I have provided numerous queries and suggestions on the annotated manuscript where important concepts were unclear.

References
Please check references for completeness and formatting. I noted at least one error in spelling an author's name and at least one incomplete reference. In the text, if citing an article you have not read, you need to indicate the source. Since you partially used Manly's symbols, I suspect that Savage 1931 might be such a case. If so, the proper text reference would be Savage (1931, cited by Manly 1993).

Figures and tables.
The captions of your figures are often incomplete and do not describe all the information. I used highlights to note some of the problems. Please check other publications or a reference on scientific writing to see what a complete figure caption should include. For example, the caption should clearly indicate which data relate to the left and right axes and which are shown by bars and lines.

The figures have some spelling mistakes on axis labels. Numbers on axes should have the same significant digits (not 0, 0.1, 0.2 . . .0.9, 1, 1.1, etc.). Style of axis labels (font, capitalization, etc.) should be consistent among the graphs. There are sometimes no spaces between words. You can use standard abbreviations for months to save space on the x-axis. The bar graphs are hard to follow because the order of food types in the key does not match the order on the graph. As noted above, lines between years should not be continuous because there was a gap in data collection.

Tables should have all items with the same number of significant digits and consistent use of 0 before decimals (unlike Table 3). Check that all abbreviations are consistent throughout the paper and use more common abbreviations to avoid readers having to remember large numbers of arbitrary expressions.

Supplementary material
The supplemental material needs to be better organized. Captions need to be improved to more completely describe the contents in acceptable English style. All supplementary material should be mentioned in the text and the numbering system within each type of item (raw data, photographs, videos) should match the order of citation in the text. You should check that all relevant data are presented in the raw data tables.

·

Basic reporting

No Comments

Experimental design

A very interesting approach

Validity of the findings

No Comments

Additional comments

No Comments

·

Basic reporting

no comments

Experimental design

no comments

Validity of the findings

no comments

Additional comments

EXCELLENT PAPER. INSIGHT INTO HABITAT NEEDS OF AN ENDANGERED SPECIES NOW THREATENED BY A PLETHORA OF HUMAN ACTIVITIES. MOST WORTHWHILE FOR PUBLICATION.

---

## Round 0.2 · Minor Revisions

Overview

This manuscript has improved considerably through revision and is now much closer to publication. Unfortunately, a number of problems remain.
• Some of the statements in the Abstract and Results do not completely agree with the data presented in the Tables.
• The Abstract presents some unnecessary and redundant information and omits some findings that appear to be more important.
• There are at least two aspects of the statistics that are unclear and possibly incorrect.
• I appreciate that the authors have had the manuscript reviewed for English language use. Unfortunately, quite a few errors remain and some wording needs to be modified to improve clarity for readers. I have provided an annotated pdf with suggested changes. (The yellow highlights show the words of concern, and the comments indicate alternative wording, punctuation, or a suggestion to delete the words.)
Details about the more substantial problems are provided below. The necessity for careful examination of the Results in relation to the Tables and for providing detailed suggestions regarding the use of English required considerable extra time from the editor. I hope careful attention to these corrections will allow the authors to have a publication-ready manuscript on the next revision and that it will help them improve the presentation of later manuscripts from their future research.

Abstract
L28. Because there is an emphasis on monthly variation in your study, you should add something about monthly variation in diet to the Abstract. For example, 'Grain consumption was lowest in November but higher from December through February. Invertebrate consumption was highest in November and February.'
L28. You should consider adding a statement about food availability to the Abstract. For example, 'Grains were most available in November and decreased through the winter, whereas invertebrates were more available in November and February than in December and January.'
L30-31. The abstract presents incorrect information on food selection. Invertebrates were not consistently preferred in December and January, and the preference for domestic crops throughout the winter applies only to grain, not to the other crops (potatoes, turnips). Present the possible causes of food selection in a separate sentence. Your wording should be more cautious. You have only a correlation, so you can suggest but you cannot conclude with certainty that the monthly changes were caused by changes in invertebrate availability. Please be extremely careful that the Abstract and Results statements agree with the data in the Tables and Figures. Not all readers will be careful enough to notice the discrepancies.

Introduction
L81, 89. To make the information as clear as possible for readers, it is important to use a consistent order of topics throughout the paper. Diet comes before availability, which comes before selection, which comes before environmental correlates.

Methods
L119-122. The list of species in the grassland is not clear. You list two species and then write 'or' followed by 6 more species. This implies that the grassland is dominated either by orchard grass or blue grass or by a combination of 6 other species, which is probably not what you mean to say. Also, remove 'etc' because a reader will have no idea what etcetera means following such a list of diverse species. If necessary, replace it with a list of the other species.
L186. Your description of sampling for food availability is still not completely clear. I assume that you placed quadrats at 100 m intervals along transects in each site. Based on this assumption, I modified your text to refer to the quadrats at the beginning of the paragraph so that readers would understand the logic of your sampling. If I am wrong, you need to correct my text and state clearly what you did.
L192. The statement about invertebrate sizes is unclear. Do you mean that when you were counting invertebrates in your samples, you only included invertebrates larger than the size of a grain because you assumed that is the minimal size that the cranes consumed? If so, you could state something such as 'We only counted invertebrates larger than the size of a single grain because that appeared to be the minimal size consumed by the cranes.' It would be better to specify 'larger than approximately X mm' rather than 'larger than the size of a single grain', if possible, because readers do not know the size of grains of the species in your study.
L196. It is not clear why you mentioned 'sampled monthly for eight months over two years' after the crops and before the grassland sampling. If that is because grasslands were sampled on a different schedule, indicate the schedule. If grasslands had the same sampling schedule, give the information after the number of quadrats for grasslands.
L215. You did not explain your methods for statistical analysis of food availability.
L220. You wrote consumed biomass when I believe you should have written available biomass. It is important to be extremely careful to avoid errors in describing your methods!
L232. It is important to explain the goal of the CCA analysis in the topic sentence before going into the detailed procedures and software used.

Results
L257. Presentation of the dietary pattern is slightly misleading. The phrase 'followed by turnips and wild plants' implies that turnips were more frequent than wild plants, which is not correct. I suggest removing the mention of turnips or stating something similar to 'Turnips comprised less than 1% of the diet'. If you refer to turnips here, do not repeat below.
L262. This statement is true only for 2013-14. I suggest replacing it with a statement similar to 'From December through February, grain consumption was more than twice as high in 2013-2014 and more than 1.4 times as high in 2014-2015.'
L265. This statement is misleading because the consumption of herbaceous plants and tubers was nearly 100 times greater than turnips. You could replace it with something similar to 'Herbaceous plants and tubers comprised less than 5% of the diet and turnips less than 0.1%, on average.'
L266. It is not clear how you did the statistics for monthly variation in diet diversity. The methods section indicates one-way ANOVA, but normally after the ANOVA shows a significant effect of month, you need a post-hoc test to determine which months showed significant differences. If you ran separate ANOVAs for each comparison of months, this would not be correct. You may need to check this with a statistician.
L282. Your diet selection information is important and the result of much effort in collecting food availability. However, your presentation is very brief and fails to provide a full synthesis of the trends. Try to indicate what you would like readers to understand from these data. Also, you must be very careful that your statements match the data in the Table.
L283-285. Your statements with reference to potatoes do not match the data in Table 3. A November preference for potatoes occurred only in 1 year, and potatoes were either avoided or showed no significant preference in the other months.
L286. This section also contains information on food availability. Does the information on environment and food availability logically precede the information on environment and food selection because you present data on availability before selection in the rest of the results?
L287. It is important to start with a topic sentence giving an overview of the findings from this analysis before giving the detailed statistical results. Readers must have a sense first of what you found in general so that they can grasp the importance of the more detailed statements of the statistical patterns.
Figure 1. I have suggested a number of corrections on the figure and caption. You have the space to make the Reserve and the China map larger within their areas. This would allow all symbols to be larger and more easily seen.
Figure 3. The caption is not clear. It refers to food use, which implies diet, but my understanding is that the analysis concerns food selection. If my interpretation is correct, I would expect a first sentence similar to the following: 'Canonical Correlation Analysis (CCA) showing the relationship between environmental variables and selection for grain, potato, and invertebrates.' [Note that I have put the food types in the order usually used in your article.] Also, I really doubt that the environmental variables should be called dependent variables. I am not very familiar with CCA, but I checked some recent articles and never saw the variables with arrows referred to as dependent. I may be wrong, but it would be a good idea to check with your statistical advisors to be sure. In addition, I think that the reference to the right and left axes is wrong; I think you mean the x-axis and y-axis (abscissa and ordinate, respectively).
Note that I have indicated a number of corrections to the table headings as well as wording for column and rows within the tables.
L626. It is important that you clarify the how you addressed the Bonferroni correction to the significance. In the heading to Table 3, you imply that the results are significant only if P<0.006 but you say non-significance occurs when P>0.05. If P is greater than .05 but less than .006, did you show it as significant or not?

Discussion
L382. You have an error in stating that cranes fed primarily on invertebrates in December and January.

---

## Round 0.3 · accepted · Accept

The manuscript has been substantially improved. I now consider it suitable for publication. Congratulations on your research and successful revisions.